

# 1 A computationally efficient statistically downscaled 100 m resolution
# 2 Greenland product from the regional climate model MAR

Marco Tedesco[1,2], Paolo Colosio[3], Xavier Fettweis[4] and Guido Cervone[5]
[1]Lamont-Doherty Earth Observatory, Columbia University, New York, 10964, USA
[2]NASA GISS, New York, 10025, USA
[3]Department of Civil, Environmental, Architectural Engineering and Mathematics, University of Brescia, Brescia, 25123, Italy
[4]Department of Geography, SPHERES research unit, University of Liège, Liège, 4000, Belgium
[5]Institute for Computational and Data Sciences and Earth and Environmental Systems Institute, The Pennsylvania State
University, University Park, PA, 16801, USA
*Correspondence to*: Marco Tedesco (mtedesco@ldeo.columbia.edu)
**Abstract.** The Greenland Ice Sheet (GrIS) has been contributing directly to sea level rise and this contribution is projected to
accelerate over next decades. A crucial tool for studying the evolution surface mass loss (e.g., surface mass balance, SMB)
consists of regional climate models (RCMs) which can provide current estimates and future projections of sea level rise
associated with such losses. However, one of the main limitations of RCMs is the relatively coarse horizontal spatial resolution
at which outputs are currently generated. Here, we report results concerning the statistical downscaling of the SMB modeled
by the Modèle Atmosphérique Régional (MAR) RCM from the original spatial resolution of 6 km to 100 m building on the
relationship between elevation and mass losses in Greenland. To this goal, we developed a geospatial framework that allows
the parallelization of the downscaling process, a crucial aspect to increase the computational efficiency of the algorithm. The
results obtained in the case of the SMB, assessed through the comparison of the modeled outputs with in-situ SMB
measurements, show a considerable improvement in the case of the downscaled product with respect to the original, coarse
output. In the case of the downscaled MAR product, the coefficient of determination ($R^2$) increases from 0.868 for the original
MAR output to 0.935 for the downscaled product. Moreover, the value of the slope and intercept of the linear regression fitting
modeled and measured SMB values shifts from 0.865 for the original MAR to 1.015 for the downscaled product in the case
of the intercept and from the value -235mm (original) to -57 mm (downscaled) in the case of the slope, considerably improving
upon results previously published in the literature.

## 26 1 Introduction

The Greenland Ice Sheet (GrIS) has been contributing directly to sea level rise since the beginning of the century
through meltwater runoff and ice mass loss. Hörhold et al. (2022) found that modern temperatures are 1.5 °C warmer than the
twentieth century and that meltwater run-off, a major contributor to sea level rise, has been consequently enhanced. Surface





melting has also been increasing since 1979, as measured by passive microwave satellite observations in terms of extension
and persistency (e.g., Tedesco et al. 2013, Colosio et al., 2021). Moreover, Hanna et al. (2021) found that over the 1972-2018
period each 1°C of summer warming corresponds to 116 Gt of surface mass loss and 26 Gt of solid ice discharge increase. A
key tool for studying the evolution surface mass loss (e.g., surface mass balance, SMB) over the GrIS is represented by (polar)
regional climate models (RCMs), which, differently from remote sensing observations can provide information on the actual
mass loss and represent an irreplaceable tool to provide future projections of such losses. A widely used model in this regard
is the Modèle Atmosphérique Régional (MAR, Fettweis et al., 2013, 2017, 2020; Tedesco et al., 2013)., a coupled surface-
atmospheric model forced at its boundaries with reanalysis data. However, one of the limitations of MAR (and of RCMs in
general) lies in the horizontal spatial resolution at which outputs can be generated. This is due to computational considerations
as well as to the physics behind the models. Currently, MAR simulations over Greenland are generated at a horizontal spatial
resolution of 6 km (e.g., Colosio et al., 2021). Such spatial resolution guarantees a sufficiently fine mesh for ice-sheet-wide
climatological studies, but it does not allow capturing fine-scale processes occurring in areas characterized by complex
topography (e.g., glaciers terminating in fjords) or small glaciated surface (e.g., ice caps). Moreover, the knowledge of mass
loss at a horizontal spatial resolution higher than the one currently available (e.g., 100s of meters) would allow to better
constrain the relationships between the surface and sub-surface and englacial hydrological systems as well as would allow a
better characterization of the meltwater fluxes into the ocean surrounding the GrIS.
To address the limitations associated to the current horizontal spatial resolution of the MAR model, statistical
downscaling can be used to enhance the spatial resolution of the modeled outputs. For example, Hanna et al. (2005, 2008,
2011) statistically downscaled reanalysis data over the GrIS. A statistical downscaling technique based on elevation correction
was also applied by Franco et al. (2012) to the 25 km MAR outputs to reconstruct GrIS SMB at 15 km spatial resolution.
Following that, Noël et al. (2016) applied an elevation dependent statistical downscaling technique to SMB components
simulated by the Regional Atmospheric Climate Model (RACMO2) at 11 km resolution to reconstruct a daily dataset of SMB
over the GrIS over a 1 km resolution grid. Here, we build upon the approach proposed by Noël et al. (2016) to generate a 100
m, statistically downscaled output of MAR SMB over the whole GrIS. Beside applying the approach to a different set of
modeled outs (MAR instead of RACMO) and the enhanced spatial resolution with respect to Noël et al. (2016), we developed
a geospatial framework that allows the parallelization of the downscaling process which increases the computational efficiency
of the algorithm. In the following, we first describe the datasets used for our approach (Section 2), then we introduce the
methodology (Section 3), followed by the results (Section 4) and our conclusions and future work (Section 5).
**2 Datasets**
**2.1 MAR Model**
Modeled quantities to be downscaled are obtained from the regional climate model MAR (Colosio et al., 2021;
Alexander et al., 2014; Fettweis et al., 2013; Fettweis et al., 2017; Tedesco et al., 2013). MAR is a modular atmospheric model



that uses the sigma-vertical coordinate to simulate airflow over complex terrain and the Soil Ice Snow Vegetation Atmosphere
Transfer scheme (SISVAT) (e.g., De Ridder and Gallée, 1998) as the surface model. The snow model in MAR, which is based
on the CROCUS model of Brun et al. (1992), calculates albedo for snow and ice as a function of snow grain properties, which
in turn depend on energy and mass fluxes within the snowpack. Lateral and lower boundary conditions are prescribed from
reanalysis datasets. Sea-surface temperature and sea-ice cover are prescribed over ocean using the same reanalysis data. The
atmospheric model within MAR interacts dynamically with SISVAT. MAR outputs have been assessed over the Greenland
ice sheet by many authors (e.g., Fettweis et al., 2017, 2020; Alexander et al., 2014).
In this study, we use the output from MAR version v3.11.5 characterized by an enhanced computational efficiency
and improved snow model parameters (Fettweis et al., 2020; Delhasse et al., 2020). The model is 6-hourly forced at the
boundaries from 1950 using ERA5 reanalysis (Hersbach et al., 2020), the newest generation of global atmospheric reanalysis
data that superseded ERA- Interim (Dee et al., 2011), and output is produced at a horizontal spatial resolution of 6 km.
Specifically, we focus our attention on daily air temperature (TT variable), surface temperature (ST variable) and surface mass
balance (SMB) outputs.

**2.2 Digital Elevation Model**

For the Digital Elevation Model (DEM), we adopt the ArcticDEM data product (Porter et al., 2018, Figure 1).
ArcticDEM is a National Geospatial-Intelligence Agency (NGA) and National Science Foundation (NSF) public-private
initiative to produce high-quality DEM of the Arctic applying stereo auto-correlation techniques to high-resolution optical
satellite images and adopting the SETSM open-source photogrammetric software (Noh and Howat, 2015). Further information
about the dataset can be found at https://www.pgc.umn.edu/guides/arcticdem/introduction-to-arcticdem/. Specifically, we use
a DEM provided at the spatial resolution of 100 m. The data are projected to the National Snow and Ice Data Center (NSIDC)
Sea Ice Polar Stereographic North and referenced to WGS84 datum. The overall dataset is composed of 403,920,000 cells and
is distributed as a GeoTIFF with a total size of approximately 1.6 Gb.

**2.3 PROMICE Surface Mass Balance measurements**

The main objective of this work is to obtain a high-resolution SMB dataset from the downscaling of the MAR model
suitable for local (i.e., glacier scale) studies. Consequently, we carried out a validation of our results by comparing the original
SMB outputs from MAR at a spatial resolution of 6 km and the downscale outputs at 100 m with in-situ SMB measurements.
For this purpose, we used the dataset collected by Machguth et al. (2016), containing 2955 measurements from 46 sites,
reported in Figure 1 as blue dots. The dataset is available on GEUS Dataverse portal (Machguth, 2022; last access 16/02/2023).
Such comprehensive dataset spans from 1892 to 2015. From the 123 years, we focused our attention to the period 1980 - 2015
when the largest portion of the dataset is temporally located and the MAR outputs are available. From the 2955 measurements
we obtained 1982 suitable SMB measurements to be used for validation. The SMB measurements are carried out by computing
the difference of stake readings between two dates. The observations are identified by the measuring site (i.e., the area or





location, containing at least one measuring point), measuring pint (i.e., specific stakes, associated with multiple readings) and the actual readings (i.e., the SMB measurement). In Table 1 we report the number of readings for each measuring site considered, together with its coordinates (WGS 84) and time period when the measurements were collected. Measurement periods are various, covering specific seasons (summer or winter SMB) or an entire year (annual SMB). In some cases, also short-term (at least one month) and multi-year measurements are present. We reconstructed the SMB in correspondence of the measurement location as algebraic sum of the daily simulated SMB between the start and end dates of the measurement. In order, as a metric to assess the performance of the downscaled product, we compute the root mean squared error (RMSE) and the least-square linear regression parameters (slope and intercept) between model outputs (SMB variable, original and downscaled) and measurements.

**2.4 GC-Net air temperature**

To test the results of the applied downscaling procedure at local scale we also compare the values of surface temperature obtained from MAR with in-situ measurements. We use data from the Greenland Climate Network (GC-Net; Steffen et al., 1996), a set of Automatic Weather Stations (AWS) located all around the Greenland ice sheet and continuously measuring air temperature, wind speed, wind direction, humidity, pressure, and other parameters. Since direct measurements of surface temperature are not available as continuous records at multiple sites around Greenland, we use the air temperature records measured at 3 m above ground level. Specifically, we consider 17 selected stations reported in Figure 1 as red triangles. Specific location and elevation for each station are also reported in Table 2 in the Results section. The AWS thermometers collect air temperature measurements at sub-daily temporal scale while MAR outputs are provided at daily temporal resolution. Consequently, we compute daily average air temperatures for the comparison with the modelled and downscaled near-surface temperatures (TT variable).

**2.5 Landsat-8 surface temperature**

As in situ measurements are only available at point scale, it is not possible to assess the potential improvement of the downscaling approach on spatially distributed fields. In the absence of spatially distributed, high spatial resolution SMB outputs, we use seven different Landsat-8 scenes covering the Jakobshavn and the Helheim Glaciers, acquired on 5 June 2015, 30 June 2015 (two images), 9 July 2015 (two images), 16 July 2015, and 18 July 2015. The Landsat-8 surface temperature product is available at 30 m spatial resolution since April 2013 and is generated from Landsat Collection 2 Level-1 thermal infrared bands and other parameters obtained from satellite observations and reanalysis data. The images were downloaded from the USGS Earth Explorer data portal (https://earthexplorer.usgs.gov/, last access 17/01/2023). We compared the Landsat-8 observations with the original and downscaled MAR outputs of surface temperature (ST variable).



## 3. Methods

### 3.1 Downscaling methodology

We adopted the methodology proposed by Noël et al. (2016) applied to the MAR regional climate model (instead of RACMO). Differently from Noël et al. (2016), however, we push the horizontal resolution of the downscaled product to 100m (instead of 1 Km). The method exploits the potential dependency of the modelled variables (e.g., surface temperature, runoff) with elevation. In order, to overcome the large number of cells and reduce the computational time, we parallelized the procedure through a combination of geospatial tools (in the software R) so that our approach can also be used for near-real time generation of downscaled maps over a specific region of the Greenland ice sheet.

The first step involves the calculation of the local dependency of the MAR outputs with respect to the elevation. For this step we refer to the methodology proposed by Noël et al. (2016). Accordingly, we compute the local linear regression (least squares) between the specific variable and the elevation (obtained from the MAR DEM) obtaining the values of slope ($m_{6km}$) and intercept ($q_{6km}$). The linear regression is carried out for each pixel of the MAR 6 km resolution DEM using the values of the adjacent pixels with a minimum of 6 points used for the regression. In the case of pixels with less than 5 adjacent pixels, we compute m and q for that pixel by interpolation. Such regression is carried out for every day and pixel of the region of interest. Figure 2 provides an example of such procedure. The local linear regression sample consists of the red pixel (5) and the surrounding green pixels (1,2,3,4,6,7,8 and 9), for a total of 9 pixels (Figure 2a). The dashed red line in Figure 2b represents the linear regression curve obtained fitting the numbered points. Parallelizing such procedure for each MAR pixel, we obtain the daily maps of $m_{6km}$ and $q_{6km}$ for the considered MAR output variable. Then, the $m_{6km}$ and $q_{6km}$ maps are reprojected to the Polar Stereographic coordinate system which is used by the DEM. The original MAR data are distributed by providing only the coordinates for the centre of each grid cell. To create a continuous grid, and avoid introducing errors, the coordinates for the four corners of each MAR grid are computed, and then they are transformed into the Polar Stereographic coordinate system. The result is a shapefile that contains a polygon for each MAR grid. Additionally, the new shapefile contains metadata to ease computations, such as a unique MAR grid ID, the Polar Stereographic coordinates for the centre of the grid, the corresponding coordinates in longitude and latitudes for the centre of the grid. The next step consists in fragmenting the high-resolution DEM into a series of smaller files, specifically one for each polygon of the reprojected MAR cells generated in the previous steps. There are a total of 55,144 files generated through each step, which are less than the total number of cells in the original MAR output. This discrepancy is due to the fact that the DEM is limited to only areas covered by the ice sheet, and it thus does not cover all the locations of where MAR output is generated. While it might seem counterintuitive that maintaining over 55,000 small files is more efficient than maintaining a single file, the answer lies in the fact that this pre-processing step enables the downscaling to be an embarrassingly parallel problem which can be efficiently solved using multi-core and multi-node infrastructure. Because the DEM is required for downscaling each grid cell, which are computed simultaneously in parallel, each task needs to read only a small file of a few kb, rather than one larger file, and it also avoids file system bottlenecks when multiple processes try accessing the same file. Most file systems do not allow for concurrent





access to the same file, and therefore if hundreds of tasks try to read the same file, each task would have to idle in a queue for
the file access to become available. This problem is prevented by generating a DEM file for each MAR grid, so that both I/O
transfer rate and file access are optimized. Furthermore, because the DEM are segmented using the original Polar Stereographic
projection, which matches the reprojected MAR grid, no further transformation is required, further speeding up the
downscaling process. The final step consists in obtaining the high-resolution maps of slope and intercept ($m_{100m}$ and $q_{100m}$) by
bilinear interpolation of  $m_{6km}$ and $q_{6km}$ over the high-resolution DEM grid. While this process was not parallelized in the
current version, it is possible to speed it up using a parallel solution.  Finally, the downscaled variable is obtained by applying
the high-resolution linear regression coefficients to the high-resolution DEM as
$$VAR_{100m} = m_{100m}H_{100m} + q_{100m}, \tag{1}$$
where VAR is the generic downscaled variable computed as linear function of high-resolution elevation of the DEM ($H_{100m}$)
through the coefficients previously obtained ($m_{100m}$ and $q_{100m}$). In Figure 2a, we show a random ensemble of points of the 100
m grid (black dots within the red pixel) around the high-resolution pixel centred in the original MAR grid (blue dot). The
values of the downscaled variable (in this example surface temperature) of such points are reported as grey dots (blue for the
central pixel) in Figure 2b, distributed along the linear regression curve. Since the origin of the MAR DEM and the high-
resolution DEM is different, errors in terms of mass conservation can arise. For example, within a MAR pixel the average
elevation of the high-resolution DEM might be higher than the original MAR elevation, possibly leading to the previously
mentioned mass conservation error (e.g., the original MAR pixel suggests for a day a lower mass loss than the ensemble of
the high-resolution pixels). For this reason, differently from Noël et al. (2016), we decided to provide physical constrains to
be satisfied as very final step of the downscaling procedure.

22        In this research, we apply the downscaling methodology to daily near-surface temperature, surface temperature and

SMB MAR outputs (Tedesco et al., 2023).

### 3.2 Spatial autocorrelation analysis and variograms

25        Beside RMSE and slop and intercept, we also focus on evaluating the potential improvements of the downscaled

product with respect to the original coarser resolution MAR outputs in terms of capability to describe the spatial distribution
of the considered variable. To this aim, we perform a spatial autocorrelation analysis using variograms. Variogram analysis is
generally adopted in geostatistical analyses to evaluate autocorrelation of spatial data (Edward et al., 1989). Autocorrelation
and variogram analysis are geostatistical tools that can be used to quantify spatial variability using metrics such as the spatial
correlation length (simply correlation length hereafter). Though these techniques were mainly designed to support the
prediction of values at locations where measurements are not available, they can be used for characterizing processes across
the scale spectrum (Herzfeld, 1993). Once process scales are known, the scale ranges over which process relationships (and





thus spatial pattern) are consistent must be determined. The knowledge of these scaling ranges will identify scales at which the process interactions change, being such scales critical for measurement or model interests (Mark and Aronson, 1984; Vedyushkin, 1994). Geostatistical methods such as spatial covariance, variogram analysis, and spectral analysis (Webster and Oliver, 2001) quantify the spatial pattern of variability of an observed property over a scale range from the minimum sample separation to the distance at which the variable becomes spatially independent. This quantified variability can, then, be used for spatial estimation based on a finite number of data points. In geostatistical approaches, spatial variation is treated as having both deterministic and stochastic components, with the deterministic component modeled using a trend surface, for example, while the stochastic component modeled as random deviations from that surface, whose spatial structure can be characterized by the variogram (Webster and Oliver, 2001). In this specific case, we fit the experimental variogram with a circular model, as it has shown to be the one that provides the highest $R^2$ when fitting the experimental data. The experimental variogram is computed as

$$\gamma(\delta) = \frac{1}{2N(\delta)} \sum_{i,j \in N(\delta)} (x_i - x_j)^2, \qquad (2)$$

where $\gamma$ is the semi-variance, $N(\delta)$ is the number of data pairs (*i-th* and *j-th*) distanced by $d$ while $x_i$ and $x_j$ are the corresponding variable values. The fitting spherical function is, then, used to compute the three main parameters characterizing the variogram: the sill, the range and the nugget effect. The sill is defined as the maximum value at which the fitted curve becomes flat; such variance value is reached at a certain distance called *range,* beyond which the data are no longer autocorrelated. The nugget corresponds to $\gamma(0)$ and it is a result of measurement errors or highly localized variability. Here, following Colosio et al. (2021), we focus our attention on the range, the descriptor of the correlation length, comparing the range values computed for the original MAR temperature outputs, the downscaled temperature and the surface temperature observed by Landsat-8.

To further investigate and quantify possible improvements in terms of spatial description of the variable of interest by th downscaled product, we also compute the so-called Structural Similarity Index Measure (SSIM). Such index has been introduced by Wang et al. (2004) to provide a similarity measure between two images. This index can objectively quantify a qualitative aspect such the similarity between two images. Considering a pair of images (X,Y) to be compared, the values assumed by the SSIM are bounded by a unique maximum (SSIM(X,Y)=1) in case X=Y, otherwise SSIM(X,Y)<1. We compute such similarity index for both original and downscaled MAR ST outputs, considering as reference the Landsat-8 surface temperature image.



## 4 Results and discussion

### 4.1 Surface and near-surface temperature

We first tested the downscaling algorithm with the MAR near-surface temperature outputs. We compared the results obtained with air temperature measurements from 17 AWS of the GC-Net. We performed the comparison by computing RMSE and $R^2$ between the modelled (original and downscaled) and the observed variable. The results obtained for the original MAR and the downscaled temperatures are reported in Table 2. Both $R^2$ and RMSE obtained for the downscaled temperatures do not exhibit significant improvements or worsening with respect to the original coarser resolution output. The difference between the 6 km and 100 m resolution is in the order $10^{-3}$ for $R^2$ and $10^{-2}$ °C for RMSE, with improvements in some stations (Swiss Camp, Crawford Pt. 1, NASA-U, Summit, Crawford Pt. 2, KAR, JAR2 and KULU) and worsening in others (Tunu-N, JAR1, South Dome and NASA-E). However, such small differences appear to be randomly distributed in space, without any clear correlation with elevation or latitude/longitude. Such results demonstrate that the applied downscaling methodology does not introduce errors in case of the TT variable at point scale.

To evaluate the results over a wider area, we considered two Landsat-8 surface temperature images collected over two different areas of the ice sheet. The two selected areas are located on the eastern and western coasts of Greenland and show a variable topography. In Figure 3 we report the surface temperature image from Landsat-8 (Figure 3a), the original ST output at 6 km spatial resolution (Figure 3b) and the downscaled ST at 100 m resolution (Figure 3c) for one of the selected Landsat-8 scenes. We compare the original MAR and downscaled high-resolution ST by computing the difference of these maps with the Landsat-8 image for the sole pixels common to all three maps. In Figure 4 where we report the histograms of the difference between Landsat-8 surface temperature and the original ST (Figure 4a) and the downscaled one (Figure 4b) for the same image. The results show no differences in terms of mean difference (μ), with an average difference of 2.7°C in both cases, similarly to the AWS comparison. Also, the standard deviation (σ) remains unvaried, being equal to 2.6 °C. Similar results have been obtained for all the compared Landsat-8 images, with mean differences ranging between -0.59°C and 3.44°C for the downscaled product (2.09°C on average) and between -0.62°C and 3.43°C for the original MAR data (2.07°C on average). We expected a similar result in terms of average difference considering the physical constrain imposed for the ST to maintain the average ST constant for each MAR pixel as final step of the downscaling procedure. These results indicate that in case of ST the downscaling algorithm does not introduce significant improvements or errors in terms of overall difference with observed temperature (expressed as RMSE for the AWS case and spatial average difference for the Landsat-8 image).

Considering such results in terms of difference at point scale and spatially averaged difference, we evaluated possible improvements in terms of spatial information content and spatial description obtained in the downscaled product. We report in Figure 5 the results of the semi-variogram analysis performed for two sub-regions of interest within the same Landsat-8 image shown in Figure 3. The two areas have been selected because of the strong differences in topography and elevation gradients. Concerning the results obtained over the topographically more complex area, we observe that the scale break of the downscaled temperature (blue line) is 13.5 km, better capturing the one from Landsat-8 data (11.5 km, red line) with respect



to the original MAR outputs (24.1 km, black line). On the other hand, the same analysis performed over an area in a more
interior region of the ice sheet, where downscaling might lead to less improvement in view of the reduced topography, does
not present improvements in terms of spatial autocorrelation (Figure 5b) and that all three datasets do not reach the semi-
variogram plateau within the considered distance. In order, to extend the comparison to another area of the ice sheet, we
performed the same variogram based analysis for another Landsat-8 scene in the surroundings of Jakobshavn glacier collected
on 11 June (Figure 6a). The map also shows the two regions of interests (ROI) selected for the analysis. We selected $ROI_1$ as
this area is characterized by a large topographic gradient within a relatively small distance and to understand the potential
improvement of the downscaling procedure over regions that are outside the main ice sheet (e.g., smaller glaciers). On the
other hand, $ROI_2$ contains both strong and mild elevation gradients (e.g., nunataks and ice sheet elevation gently increasing as
moving towards the interior). This area is covered by most of its portion containing the ice sheet (right portion of $ROI_2$) by the
DEM, being this absent in the case of the left portion of the ROI, where fjords and ocean features are dominating. In case of
$ROI_1$ (Figure 6b), the variogram analysis indicates that the break scale distance for Landsat-8 when considering only the pixels
where the DEM is available is 7.5 km. This value becomes 14.6 km for the high-resolution map of ST and 24.7 km in the case
of the original MAR outputs, suggesting that the downscaled product is able to perform better than the original one in terms
of spatial scale similarity with respect to the Landsat-8 data. The mean difference between Landsat-8 and the downscaled
(original MAR) surface temperature, considering only the pixels where the DEM is available, are 1.69 °C (1.7 °C) with a
standard deviation of 2.02 °C (2.14 °C), with differences of the same order of magnitude obtained in the previous analysis for
the other Landsat-8 image. When considering all pixels within the ROI (e.g., also where no DEM is available), the mean
differences between Landsat-8 and downscaled (original) MAR surface temperature become 1.89 °C (2.12 °C) with a standard
deviation of 2.15 °C (2.23 °C). In this case, the scale breaks for the original and the downscaled MAR versions are similar, ~
25 km (~ 16km in the case of Landsat-8). We point out that the scale breaks are sensitive to the different physical processes
driving the spatial properties. The $ROI_2$ contains both strong and mild elevation gradients given the presence of nunataks and
the slow ice sheet increasing elevation after the ice cliff begins. The area is covered by most of its portion containing the ice
sheet (right of the image) by the DEM, which, however, is absent in the case of the left portion of the ROI, containing fjords
and the ocean. The scale breaks for the Landsat-8, downscaled and original MAR cases for the portion of the $ROI_2$ where the
DEM is available are close to each other, on the order of ~ 25 km. We observe an improvement in the SSIM in the case of the
downscaled data by 30 % (from 0.33 in the case of the original MAR resolution to 0.43 in the case of downscaled MAR).
Also, despite as expected the mean and standard deviation of the distribution of the differences between the Landsat-8 data
and the simulated quantities remains similar, we notice a reduction in both the mean (from 0.86°C for original MAR to 0.83°C
for the downscaled product) and of the standard deviation (from 0.71°C for original MAR to 0.63°C for the downscaled
product) when downscaling the MAR output. We further note that when considering all pixels (including those where no
DEM), the SSIM of the two products improves from 0.11 (original) and 0.14 (downscaled) and that the scale break of the
original MAR products is larger (~ 63km) than the one of the Landsat-8 data (~ 21 km). In synthesis, the downscaling does
not introduce any considerable bias on the original value, preserves the total integrated quantity of energy within each area





and improves, from a quantitatively point of view, the spatial performance of the MAR outputs by generating a product that
has a spatial structure that is closer to the one of the observed remote sensing dataset.
**4.2 Surface Mass Balance**
After applying the downscaling algorithm to surface temperature, we applied it to MAR SMB outputs of SMB and
assessed the results obtained with in situ measurements from the dataset collected by Machguth et al. (2016). As mentioned,
we compared 1982 SMB measurements carried out between 1980 and 2015 and localized in the ablation area of the GrIS
(Table 1). Figure 7 shows the scatterplots of the comparison of modelled SMB from the original MAR (Figure 7a) and the
downscaled product (Figure 7b) with in situ measured SMB. Our results show that the downscaled product better estimates
the measured SMB, exhibiting an increased $R^2$ from an already relatively high value of 0.868 for the original MAR to 0.935
for the downscaled product. As a comparison, Noël et al. (2016) obtained an increase of $R^2$ from the downscaling of SMB
outputs of the RACMO regional climate model from 0.47 in the case of the original 11 km spatial resolution outputs to 0.78
in case of the downscaled SMB (1 km resolution). We point out that, in our case, the starting value of $R^2$ for the original MAR
product already exceeds the value obtained in the case of the downscaled RACMO outputs.
The values of slope and intercept of the best-fitting line improve as well when considering the downscaled product.
The value of the slope shifts from 0.865 for the original MAR to 1.015 for the downscaled product; similarly, the intercept
increases from the value -235 mm of the coarse resolution outputs to -57 mm of the downscaled SMB, closer to its optimal
value (i.e., null intercept). As a comparison, the downscaling algorithm of Noël et al. (2016) applied to RACMO improved the
estimate of SMB in terms slope from 0.72 to 1.03, with a slight increase of the intercept (from 70 mm to 100 mm). The RMSE
between modelled and measured SMB also decreases in the case of the downscaled product from 669 mm for the 6 km outputs
to 511 mm for the 100 m case, significantly improving the estimate of SMB at local scale. Noël et al. (2016) obtained a
reduction of the RMSE passing from a value of 1200 mm for the 11 km RACMO outputs to a value of 740 mm for the 1 km
case. Fettweis et al. (2020) compared MAR and RACMO, among 13 models of four types (positive degree day models, energy
balance models, regional climate models and general circulation models), SMB estimates with the same PROMICE in situ
measurements within the GrIS SMB model intercomparison project (GrSMBMIP). They considered only the measurements
collected between 1980 and 2012 and with measurement period longer than 3 months. They also excluded the records located
outside the 1 km ice mask they used for the intercomparison of the models, for a total of 1438 SMB measurements. The model
versions in this case are MARv3.9.6, an older version than the one we adopted and at the spatial resolution of 15 km, and
RACMO2.3p2 (Noël et al., 2019), a new version of the one adopted in Noël et al. (2016) and with a spatial resolution of 5.5
km. From the comparison, they obtained a RMSE of 480 mm for MAR and 630 mm for RACMO. In both cases, the RMSE is
lower than the one obtained in this work for MAR (both original and downscaled) and by Noël et al. (2016) for RACMO. The
difference can be related to the differences in spatial resolution and model versions and, most probably, to the sub-sampling
of the SMB measurements discarding short-term records (i.e., measurement period lower than 3 months), suggesting that the
bias might be dissipated for longer observation periods.





To further investigate our results, we compute the variation in RMSE between the 6 km spatial resolution MAR
outputs and the downscaled product for different elevation classes, longitude and latitude ranges and for each specific
glacier/location (i.e., for each station ID, Table 1) of the PROMICE in situ SMB dataset. The RMSE difference is computed
as $\Delta RMSE=RMSE_{6km}-RMSE_{100m}$ (i.e., improvements are characterized by negative values of $\Delta RMSE$) and the results obtained
are reported in Figure 8 grouped by location (Figure 8a), elevation (Figure 8b), latitude (Figure 8c) and longitude (Figure 8d).
The results exhibit improvements in the estimate of SMB at all the altitudes besides the 100-200 m asl, 200-300 m asl and
1300-1400 m asl elevation classes, with the best performance obtained at 700-800 m asl and 800-900 m asl elevation classes.
The results grouped by latitudinal bands show the highest improvements in south Greenland; a decrease in performance has
been recorded in the latitudinal band 67.5-70 °N where the only Paakitsoq JAR ($\Delta RMSE= 181$ mm, worst result obtained) and
Swiss Camp/ST2 ($\Delta RMSE= -127$ mm) measurement sites are located, and the improvement obtained in case of Swiss Camp
is strongly counterbalanced by the reduced performances in Paakitsoq JAR. However, the longitudinal classes do not present
any decrease of the performances, indicating that the worsening in the spotted critical stations is counterbalanced by the
improvements measured in the others. We obtained a decrease in performances in 6 out of 22 considered cases with the worst
result obtained for the already presented Paakitsoq JAR case. In the 5 other cases (i.e., Hans Tausen Ice Cap,
Nioghalfvjerdsfjorden, Isertoq, Nordbo Glacier and K-Transect) we recorded an average $\Delta RMSE$ of 26 mm (ranging from 6
mm to 80 mm). On the other hand, we obtained an improvement in 16 out of 22 measurement sites with the best performances
in case of A.P. Olsen Ice Cap ($\Delta RMSE= -611$ mm). In the other 15 cases (i.e., Qaanaaq Ice Cap, Petermann, Hare Glacier,
Kronprins Chistian Land, Storstrømmen, Freya Glacier, Violin Glacier, Helheim, Isortuarssup Sermia, Qamanarssup Sermia,
Kangilinnguata Sermia, Qapiarfiup, Amitsuloq Ice Cap, Tasersiaq and Swiss Camp/ST2) we found an average decrease in
RMSE of 183 mm (ranging between 59 mm and 371 mm). Even if such reduction of performances in terms of SMB estimate
accounts for 27% of the considered stations, it does not compromise the overall improvement, being smaller in terms of
average, minimum and maximum absolute values of $\Delta RMSE$ than the results obtained for the stations were improvement
occurred.
**5 Conclusions and future work**
We applied a statistical downscaling technique to increase the horizontal spatial resolution of the outputs of the MAR
regional climate model from 6 km to 100 m for the surface temperature and SMB quantities. The approach builds on the
dependency of such quantities on elevation, as originally proposed in Noël et al. (2016). Here, however, the technique was
applied to the output of a different climate model (RACMO) and the spatial resolution of the downscaled product was 1 km,
rather than 100 m. Moreover, differently from Noël et al. (2016), we imposed a mass conservation so that the overall mass
obtained for each pixel at high resolution nested within a coarse resolution one is preserved. To address the computational
issues associated with the relatively high spatial resolution, we developed a geospatial, parallelized framework that allows to
perform the downscaling over the whole ice sheet in an efficient way.





We, first, tested our approach by applying it to surface temperature data and assess the outputs using both in-situ ad
satellite data. Our results showed no significant improvement nor deterioration of the downscaled product with respect to the
original MAR outputs. This confirms that our approach was not introducing any bias and was properly implemented. However,
we found improvement of the downscaled surface temperature when analyzing the skills of the downscaled product to capture
the spatial scales (e.g., scale breaks) of the observed surface temperature fields. The results obtained in the case of the SMB,
assessed through the comparison of the modeled outputs with in-situ SMB measurements, show a considerable improvement
in the case of the downscaled product with respect to the original, coarse output, evaluated through the analysis of the RMSE,
$R^2$ and the slope and intercept of the modeled vs. measured values. In the case of the downscaled MAR product, the $R^2$ value
increases from 0.868 for the original MAR to 0.935 for the downscaled product with the value of the slope and intercept
shifting from 0.865 for the original MAR to 1.015 for the downscaled product in the case of the intercept and from the value
-235mm of the coarse resolution outputs to -57 mm of the downscaled SMB in the case of the slope. As a reference, Noël et
al. (2016) obtained an increase of $R^2$ from the downscaling of SMB outputs of the RACMO regional climate model from 0.47
in the case of the original 11 km spatial resolution outputs to 0.78 in case of the downscaled SMB (1 km resolution) and a shift
in the slope and intercept from 0.72 to 1.03 (slope) and from70 mm to 100 mm (intercept). An analysis of the performance of
the downscaled product for different elevation classes, longitude, and latitude ranges and for each specific glacier/location
where SMB in-situ data is available shows that for 27% of the stations, the downscaled product does not perform as expected.
However, this does not compromise the overall improvement, being the deterioration of the performance smaller in terms of
average, minimum and maximum absolute values of ΔRMSE than the results obtained for the stations were improvement
occurred.
The next step is to implement a similar approach for downscaling MAR outputs over both the Greenland and
Antarctica ice sheet using machine learning (ML) based approaches. Indeed, the approach here proposed cannot be easily
extended to Antarctica, where surface melting does not exhibit a strong dependency from elevation, as most of it occurs over
ice shelves, at the sea level and where little elevation gradients exist. Moreover, improvements to the downscaling of the SMB
can be obtained by either considering complementary inputs that can improve estimates of losses (e.g., remotely sensed albedo)
or of mass gains (e.g., accumulation). ML tools can help in this regard. ML tools have, indeed, been used for improving
predictions beyond that of state-of-the-art physical models or for improving parameterization in models. In particular,
conditional generative adversarial networks (C-GANs or simply GANs in the following) can be successfully applied to the
problems discussed above (Goodfellow et al., 2014). GANs is a class of ML tools in which two neural networks compete with
each other in a min-max optimization problem. The first network, called generator, aims to generate new data samples that are
indistinguishable from the training data (e.g., high-resolution melting maps obtained from the remote sensing observations)
by the other network, called discriminator. In our case the GAN aims to generate high-resolution melting maps that are
indistinguishable by the second network from high-resolution remote sensing observations or numerical model outputs. We
have already started to build the architecture for this framework and are in the phase of collecting the necessary datasets and
build the proper data framework to perform such work.





## Code and data availability

The MAR v3.11.5 code and outputs are available at https://www.mar.cnrs.fr/ and ftp://climato.be/fettweis/MARv3.11. Automatic whether station data are available on EnviDat portal (https://www.envidat.ch/#/metadata/gcnet, Steffen et al., 2020, last access 16/02/2023). Surface mass balance measurements are available on GEUS Dataverse portal (https://doi.org/10.22008/FK2/5VNBQA, Machguth, 2015, last access 16/02/2023). Landsat-8 images are available on the USGS Earth Explorer portal (https://earthexplorer.usgs.gov/, last access 16/02/2023). Downscaled data is available at https://doi.org/10.5281/zenodo.7803611. Downscaling code is available upon request to mtedesco@ldeo.columbia.edu.

## Authors contributions

MT, PC and GC designed the study and the methodology. GC optimized the algorithm parallelization. PC and MT performed the comparison with measured and remotely sensed observations and wrote the first draft of the paper. XF run the MARv3.11.5 model and provided the outputs. All the authors discussed the results and contributed to the paper.

## Competing interests

Xavier Fettweis is a member of the editorial board of The Cryosphere,

## Acknowledgment

This work was supported by the NASA grant 80NSSC17K0351, NSF grants OPP-1713072 and OPP-2136938 and the Heising-Simons Foundation grant HSFOUND 2019-1160. PC would like to thank Lamont-Doherty Earth Observatory and University of Brescia for the support provided.

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

Figure 1: Map of Greenland ice sheet. The digital elevation model (DEM) at 100 m resolution is represented in greyscale, the GC-Net air temperature locations are plotted as red triangles and the PROMICE surface mass balance measurements locations are reported as blue dots. The two rectangles indicate the Jakobshavn (blue) and Helheim (black) regions.

2
3
4



| ID | Glacier/Site name | Latitude [°] | Longitude [°] | Measurement years | Points | Readings |
|---|---|---|---|---|---|---|
| 126 | Qaanaaq Ice cap | 77° 30' 36'' N | 69° 9' 0'' W | 2012-2015 | 6 | 12 |
| 128 | Petermann | 80° 41' 2'' N | 60° 17' 35'' W | 2002-2013 | 2 | 4 |
| 130 | Hans Tausen Ice Cap | 82° 29' 24'' | 37° 30' 0'' W | 1995 and earlier | 5 | 13 |
| 140 | Hare Glacier | 82° 50' 24'' N | 36° 40' 12'' W | 1994-95 | 29 | 62 |
| 170 | Kronprins Christian Land | 79° 46' 48'' N | 25° 11' 24'' W | 1993-1994, 2008-2013 | 20 | 62 |
| 180 | Nioghalfvjerdsfjorden | 79° 30' 0'' N | 21° 36' 0'' W | 1996-1997 | 13 | 13 |
| 215 | Storstrømmen | 77° 30' 0'' N | 23° 0' 0'' W | 1989-1994 | 22 | 113 |
| 220 | A.P. Olsen Ice Cap | 74° 38' 24'' N | 21° 26' 60'' W | 2008-2013 | 17 | 56 |
| 230 | Freya Glacier | 74° 22' 48'' N | 20° 49' 12'' W | 2008-2013 | 29 | 93 |
| 232 | Violin Glacier | 72° 20' 60'' N | 26° 58' 48'' W | 2008-2013 | 2 | 12 |
| 254 | Helheim | 66° 24' 36'' N | 38° 20' 24'' W | 2008-2010 | 21 | 118 |
| 270 | Isertoq | 65° 42' 0'' N | 38° 53' 24'' W | 2007-2013 | 2 | 15 |
| 315 | Nordbo Glacier | 61° 30' 0'' N | 45° 22' 12'' W | 1977-83 | 41 | 200 |
| 412 | Isortuarssup Sermia | 63° 47' 60'' N | 49° 47' 60'' W | 1983-1988 | 3 | 9 |
| 414 | Qamanarssup Sermia | 64° 30' 0'' N | 49° 23' 60'' W | 1979-1988, 2007-2013 | 20 | 164 |
| 416 | Kangilinnguata Sermia | 64° 52' 48'' N | 49° 17' 60'' W | 2010-2013 | 1 | 3 |
| 420 | Qapiarfiup | 65° 34' 48'' N | 52° 12' 36'' W | 1980-1989 | 5 | 75 |
| 440 | Amitsuloq Ice Cap | 66° 8' 24'' N | 50° 19' 12'' W | 1981-1990 | 26 | 422 |
| 450 | Tasersiaq | 66° 15' 36'' N | 51° 23' 60'' W | 1982-1989 | 6 | 111 |
| 454 | K-Transect | 67° 5' 60'' N | 48° 51' 36'' W | 1990-2013 | 11 | 193 |
| 456 | Paakitsoq, JAR | 69° 29' 24'' N | 49°51' 36'' W | 1982-1992, 1996-2013 | 22 | 220 |





| 458 | Swiss Camp/ST2 | 69° 33' 53'' N | 49° 19' 51'' W | 1990-2014 | 2 | 12 |
|---|---|---|---|---|---|---|

2 **Table 1: PROMICE surface mass balance measurements information for the selected Glaciers and measurements sites.**

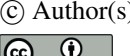
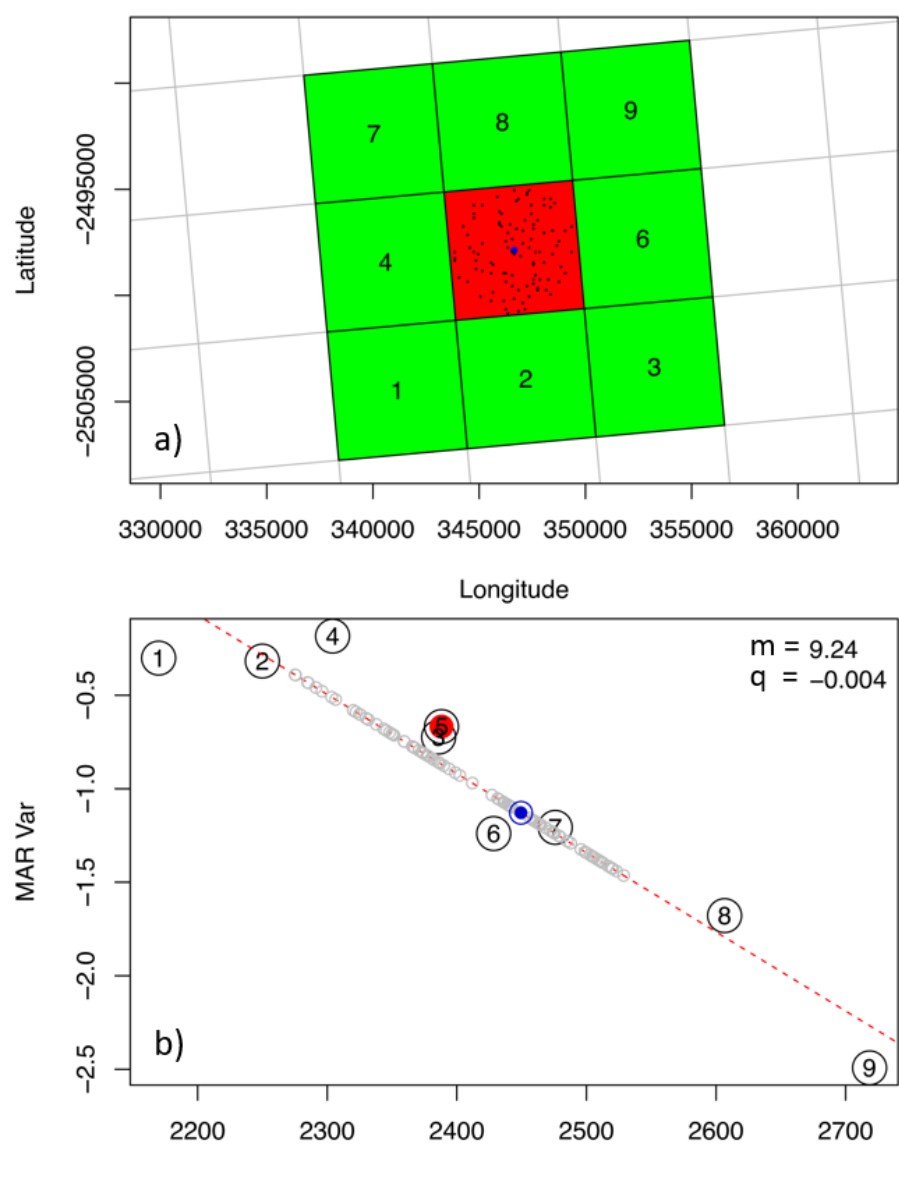

**Figure 2: Elevation downscaling procedure example for a generic variable. In panel (a) the considered MAR pixel (red) and the surrounding pixels (green) adopted for the local linear regression are represented. In panel (b) the variable value of each considered pixel is reported as numbered circle. The dashed red line represents the linear regression computed for such pixels and the grey circles represent the downscaled variable for a group of 100 m pixels randomly picked within the considered MAR pixel.**



| Station | Latitude [°] | Longitude [°] | Elevation [m] | $R_2$ MAR$_{6km}$ | $R_2$ MAR$_{100m}$ | RMSE MAR$_{6km}$ | RMSE MAR$_{100m}$ |
|---|---|---|---|---|---|---|---|
| Swiss Camp | 69° 34' 06" N | 49° 18' 57" W | 1149 | 0.945 | 0.945 | 2.37 | 2.36 |
| Crawford Pt.1 | 69° 52' 47" N | 46° 59' 12" W | 2022 | 0.872 | 0.873 | 3.95 | 3.95 |
| NASA-U | 73° 50' 31" N | 49° 29' 54" W | 2369 | 0.788 | 0.789 | 5.35 | 5.34 |
| GITS | 77° 08' 16" N, | 61° 02' 28" W | 1887 | 0.915 | 0.915 | 3.4 | 3.4 |
| Humboldt | 78° 31' 36" N | 56° 49' 50" W | 1995 | 0.801 | 0.801 | 5.64 | 5.64 |
| Summit | 72° 34' 47" N | 38° 30' 16" W | 3254 | 0.837 | 0.84 | 4.62 | 4.58 |
| Tunu-N | 78° 01' 0" N | 33° 59' 38" W | 2113 | 0.937 | 0.936 | 3.17 | 3.2 |
| DYE2 | 66° 28' 48" N | 46° 16' 44" W | 2165 | 0.94 | 0.94 | 2.72 | 2.72 |
| JAR1 | 69° 29' 54" N | 49° 40' 54" W | 962 | 0.787 | 0.786 | 4.37 | 4.38 |
| Saddle | 66° 00' 02" N | 44° 30' 05" W | 2559 | 0.935 | 0.935 | 2.77 | 2.77 |
| South Dome | 63° 08' 56" N | 44° 49' 00" W | 2922 | 0.915 | 0.915 | 2.76 | 2.77 |
| NASA-E | 75° 00' 00" N | 29° 59' 59" W | 2631 | 0.882 | 0.881 | 3.94 | 3.97 |
| Crawford Pt.2 | 69° 54' 48" N | 46° 51' 17" W | 1990 | 0.893 | 0.894 | 3.62 | 3.61 |
| NASA-SE | 66° 28' 47" N | 42°30' 00" W | 2425 | 0.86 | 0.86 | 3.83 | 3.83 |
| KAR | 69° 41' 58" N | 33° 00' 21" W | 2579 | 0.935 | 0.936 | 2.6 | 2.57 |
| JAR2 | 69º 25' 12" N | 50º 03' 27" W | 568 | 0.706 | 0.709 | 4.79 | 4.76 |
| KULU | 65° 45' 30" N | 39° 36' 06" W | 878 | 0.59 | 0.595 | 5.22 | 5.19 |

2 **Table 2: Root-mean-square error and $R^2$ computed comparing MAR$_{6km}$ and MAR$_{100m}$ with air temperature measurements from the**
3 **GC-Net considered stations. Longitude, latitude and elevation of the station are also reported.**



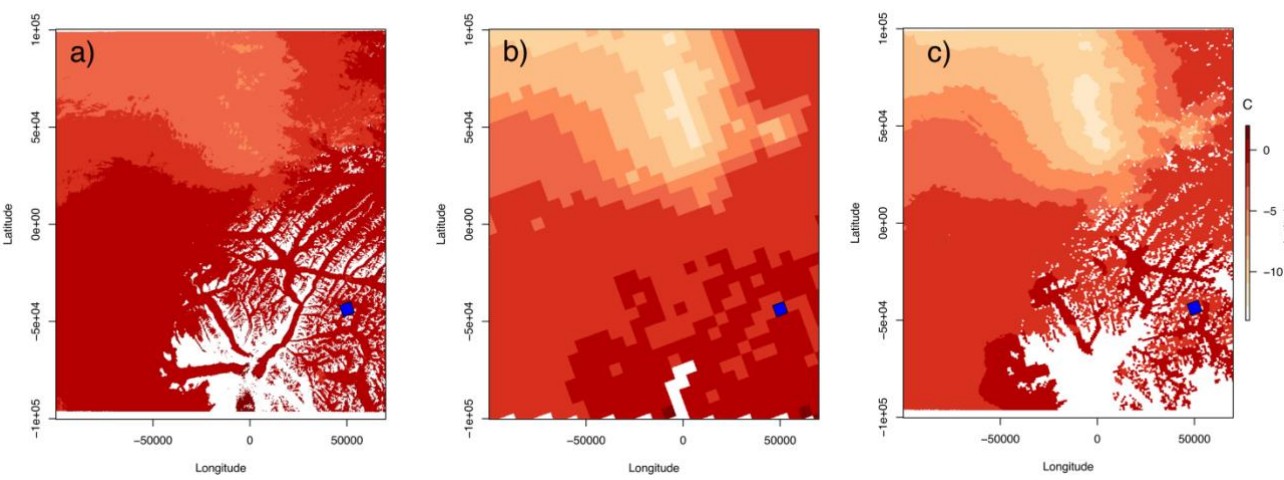

**Figure 3: Maps of temperature from (a) Landsat-8, (b) MAR$_{6km}$v2 and (c) MAR$_{100m}$ over the area covered by the Landsat 8 selected image on 30 June 2015. The blue dot reported to every map represents the 6 km pixel of the original MAR grid reported in red in Figure 2a.**





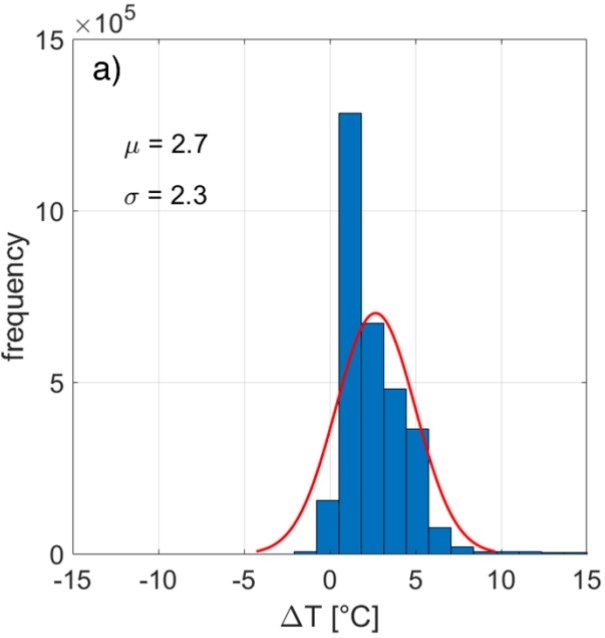
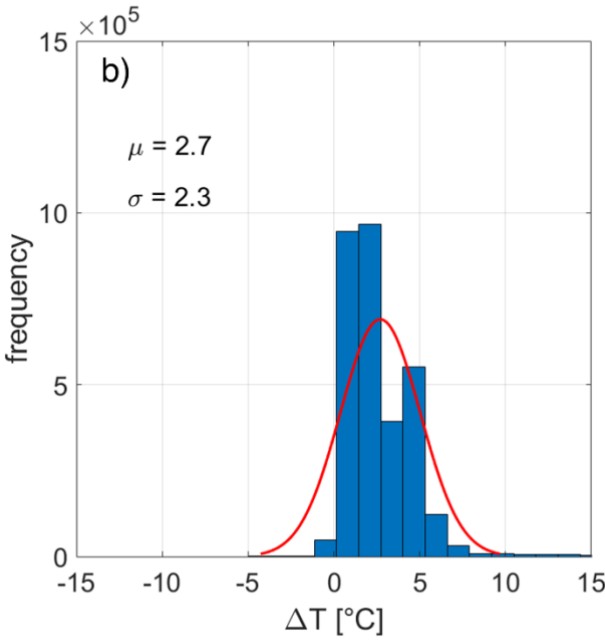

2  **Figure 4: Histograms of the difference (a) between the 6 km MAR temperature and Landsat-8 temperature and (b) between 100 m**
3  **MAR temperature and Landsat-8 temperature.**



3
4  **Figure 5: Modelled semi variograms for the Landsat-8, MAR$_{6km}$ and MAR$_{100m}$ computed over two regions of interest reported in**
5  **the inset.**



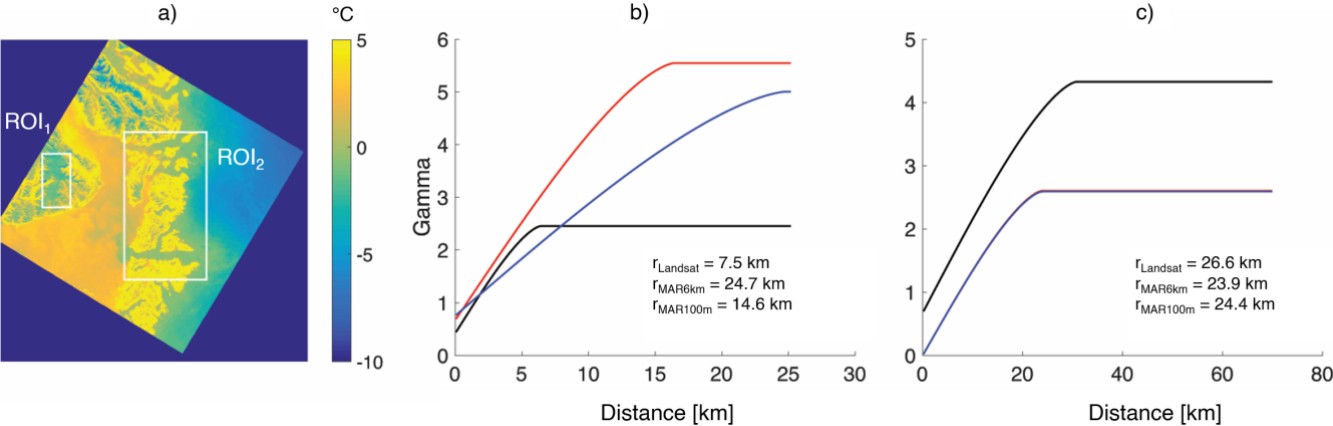

Figure 6: (a) Landsat-8 temperature captured on 11 June 2015 over areas around the Jakobshavn Glacier and (b, c) modelled semi variograms for the Landsat-8, MAR$_{6km}$ and MAR$_{100m}$ computed over (b) the first region of interest (ROI$_1$) and (b) the second region of interest (ROI$_2$).





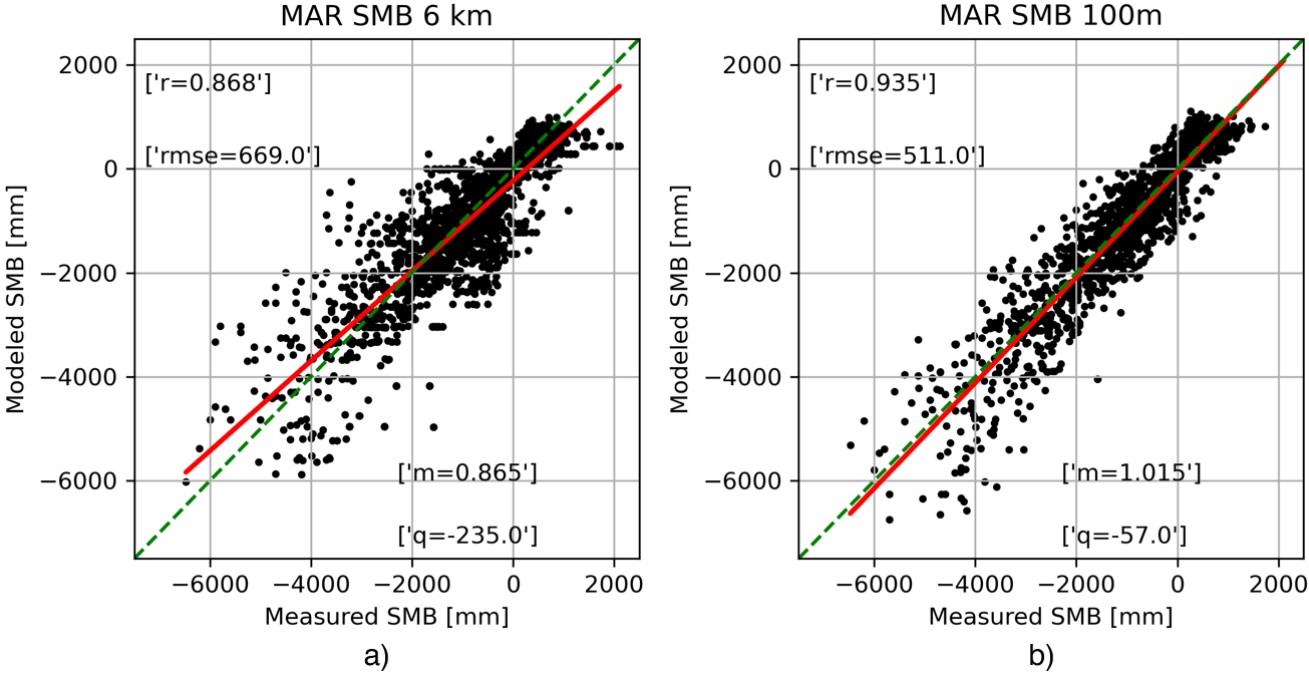

2 **Figure 7: Comparison between measured and modelled surface mass balance from (a) original 6 km MAR and (b) downscaled 100**
3 **m MAR.**





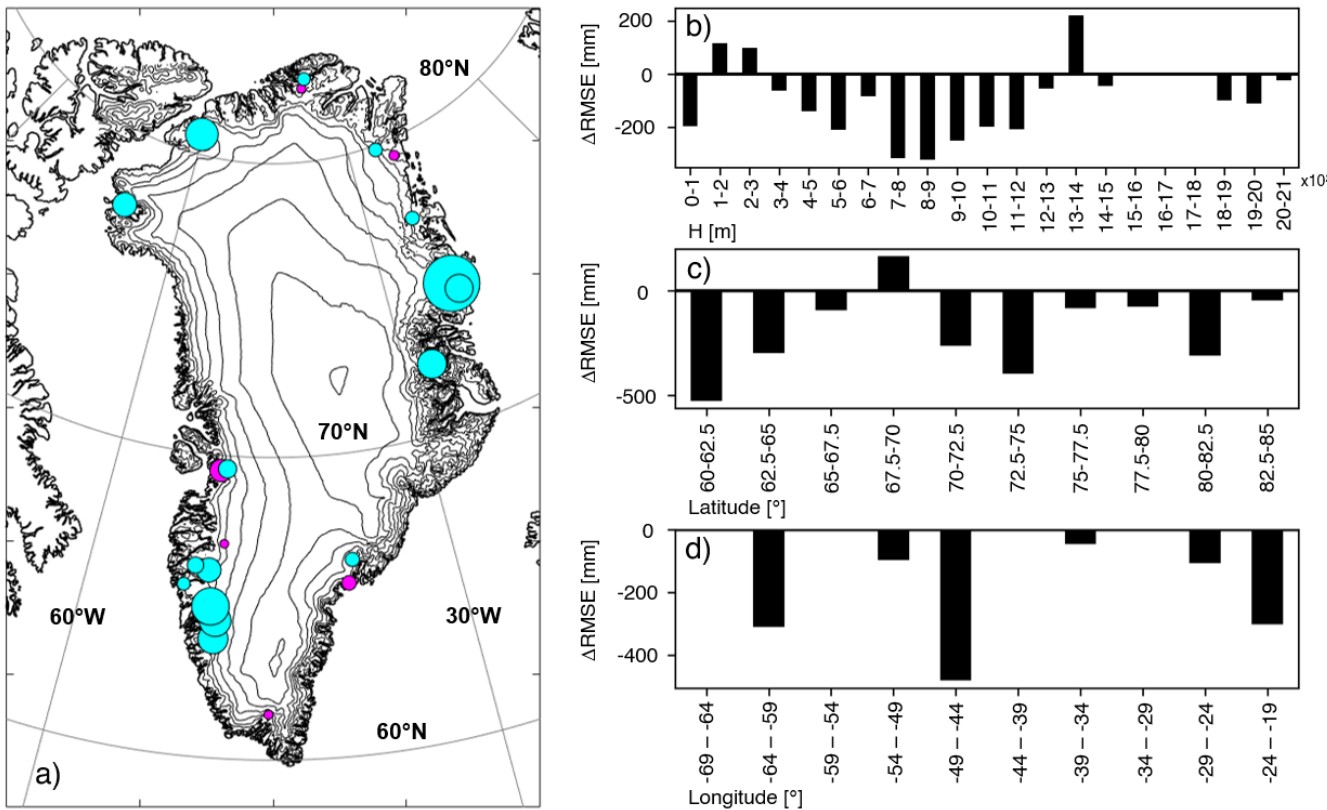

**Figure 8: Difference between original and downscaled MAR modelled surface mass balance RMSE with respect to the measured surface mass balance data ($RMSE_{100m}$-$RMSE_{6km}$) by (a) glacier, (b) elevation, (c) latitude and (d) longitude. In the bubble chart map the contour lines are plotted every 500 m (original $MAR_{6\,km}$ DEM), positive values (worsening) of ΔRMSE are reported in magenta while negative values (improvement) in cyan.**