# Peer review of "A computationally efficient statistically downscaled 100 m resolution"

_The Cryosphere, 2023_

## Referee Comment (RC1)

**1 General comments**

This manuscript presents a Greenland Surface Mass Balance (SMB) product which was downscaled from 6km to 100m resolution using output from the regional climate model MAR and demonstrates that the downscaled dataset exhibits an predominantly better agreement with observations than the respective original MAR output at its native resolution (which is already
5 at a very high resolution). To my knowledge the data product is unique in its extremely high resolution and Greenland wide coverage. The presented analysis convincingly demonstrates the improved quality of the SMB data and this work could be a valuable source for the community with respect to small scale applications. The manuscript is clearly structured and most parts are easy to understand, even though some sentences could possibly be decluttered and shortened (examples in the specific comments).
10 Nevertheless, being interested in downscaling approaches in general rather than in small scale applications, I have some major concerns which mostly concern the general approach.

**2 Major comments**

The downscaling approach will be most effective where the MAR topography and the 100m DEM strongly differ and where topographic gradients are large and are dominating the temperature distribution. Towards the coast and on high altitude plateaus
15 the temperature and SMB distribution might be unrelated to elevation. Please provide a map of height difference between the 100m DEM and the native MAR orography, possibly in a supplement. Similarly, I propose to include the height at in situ location, and respective heights according to the DEM and MAR orography in table 1 and 2. Also a map of the topographic slope might be helpful.

20 It is not clearly stated, and it should be, where the here applied downscaling approach differs from the one in Noel et al. (2016). An indeed major difference is, that here SMB is downscaled directly (p. 6, l.23), while in Noel et al. (2016) only the SMB components melt, runoff and sublimation are downscaled while precipitation is interpolated and SMB and refreezing are recalculated from the downscaled components. I am not convinced that downscaling SMB in total is a similarly good choice and would be interested to see the correlation of SMB to elevation (similar to Fig.3 in Noel et al. (2016)). Furthermore it should
25 be explained how grid points outside of the 6km ice mask are treated.

I also wonder how much information is actually gained from going to ever increasing resolution (e.g. when going from 6km to 1km to 100m). Is it possible to repeat the SMB downscaling for 1km and compare to stake measurements?

30 Where the correlation of a variable to elevation is weak, an elevation based downscaling will likely smooth the signal rather than adding finer structure (since regression parameters are interpolated). In these regions I would expect that simple interpolation to 100m resolution would produce better results. Therefore it would be interesting to see the same statistics for 6km-SMBs being interpolated to the precise stake location.

35 # 3 Specific comments

p. 1, l 21: „In the case of the downscaled MAR product": unnecessary repetition. The formulation "in the case of" is heavily used in general and in some case it is redundant or makes the text a bit clumsy.
p. 1, ll. 23-24: slope and intercept are interchanged.
p. 1, l. 28: specify that this study was analyzing North and Central Greenland
40 p.2, l.14: maybe provide references for datasets which provide resolution of 100s of meters.
p.2, ll. 15-16: this is a bit elusive. Can you specify how understanding englacial systems or ocean interaction would benefit from higher degree of detail at the surface (given that mass is conserved with respect to the source data)?
p.3,l.12: is the TT variable 3-dimensional air temperature or near surface air temperature? Is it possible to specify the height

above surface?

p.4,l.1: typo, pint-> point

p.4,l.11: precise: values of near surface air temperature

p.4,l.24: . . . we use surface temperature fields from seven different. . .

5  p.5,l.13-14: It should be stated that this (I guess) is referring to pixels at the margins of the ice masks

p.6,l.20: specify what the physical constraints are in terms of temperature and SMB.

p.6,l.21: typo, constrains – constraints?

p.6,l.25: typo, slop->slope

section 3.2: I had a hard time reading this section. Maybe concentrate a bit more on what information this analysis provides

10  and how to interpret it. Introduce scale brakes here.

p.8,l.5: what do you use for the comparison with the original MAR output? Is it nearest neighbor or do you interpolate to the station location or do you interpolate to 100m grid and then choose the nearest neighbor?

p.8,l.18: this is unclear to me. What are the sole pixels?

p.8,l.24: reword, maybe: the similarity in mean differences is not surprising. . .

15  p.9, ll.10-12: confusing sentence. Please rephrase.

p.9, ll.28-31: confusing, please rephrase. Maybe: Against our expection. . .

p.10, ll.31-33: It needs to be noted that Fettweis et al. (2020) also applied an elevation correction and interpolated to in stake location.

20  Figures: please check x and y labels for the maps (distance, longitude, latitude)

Fig. 2: m and q are not consistent with Eq. 1 (interchanged)

Fig. 3: check colorbar label

Fig. 8: it would be interesting to also show RMSE by topographic slope.

**References**

Fettweis, X., Hofer, S., Krebs-Kanzow, U., Amory, C., Aoki, T., Berends, C. J., Born, A., Box, J. E., Delhasse, A., Fujita, K., Gierz, P., Goelzer, H., Hanna, E., Hashimoto, A., Huybrechts, P., Kapsch, M.-L., King, M. D., Kittel, C., Lang, C., Langen, P. L., Lenaerts, J. T. M., Liston, G. E., Lohmann, G., Mernild, S. H., Mikolajewicz, U., Modali, K., Mottram, R. H., Niwano, M., Noël, B., Ryan, J. C., Smith, A., Streffing, J., Tedesco, M., van de Berg, W. J., van den Broeke, M., van de Wal, R. S. W., van Kampenhout, L., Wilton, D., Wouters, B., Ziemen, F., and Zolles, T.: GrSMBMIP: intercomparison of the modelled 1980–2012 surface mass balance over the Greenland Ice Sheet, The Cryosphere, 14, 3935–3958, https://doi.org/10.5194/tc-14-3935-2020, https://tc.copernicus.org/articles/14/3935/2020/, 2020.

Noel, B., van de Berg, W. J., Machguth, H., Lhermitte, S., Howat, I., Fettweis, X., and van den Broeke, M. R.: A daily, 1 km resolution data set of downscaled Greenland ice sheet surface mass balance (1958-2015), CRYOSPHERE, 10, 2361–2377, https://doi.org/10.5194/tc-10-2361-2016, 2016.

---

## Referee Comment (RC2)

**Manuscript title:** A computationally efficient statistically downscaled 100 m resolution Greenland product from the regional climate model MAR

**Authors:** Marco Tedesco, Paolo Colosio, Xavier Fettweis, Guido Cervone
* * *
**General comments**

This study uses a statistical downscaling technique to enhance the horizontal resolution of the Modèle Atmosphérique Régional (MAR) regional climate model and produce an improved surface mass balance (SMB) product. The authors use an impressive breadth of datasets and tools both for statistically downscaling and for evaluating the final product. The product itself is clearly an improvement, is of notably high spatial resolution, and likely has many valuable applications. The high resolution over such a large area is computationally remarkable. The manuscript is well-organized and the flow of ideas is very logical.

My main concern is the final step of the methodology, which leaves the reader wondering how well the statistical downscaling works without applying physical constraints. I have also described two minor comments and several line-by-line comments that are mostly concerned with improving the writing itself. Some clarity of the research is lost due to longer/confusing sentences, so I have suggested some improvements below.

**Major comments**

Description of final step of methodology: There is an insufficient description of the "physical constraints" applied in the final step of the downscaling in section 3.1 This needs further elaboration, especially since it differs from Noël et al. (2016) and is later referred to in section 4.1. The authors should (1) provide a reason as to why the mass conservation issues did not arise in Noël et al. (2016), (2) describe exactly what these physical constraints are, and (3) report on how they affect the final product. Without any additional information, the application of the physical constraints could be interpreted as forcing the final product to fit within the expectations. As this is likely not the case, a description of these steps will give the reader more confidence in the methodology.

**Minor comments**

Scale break: I am unsure of the meaning of "scale break" (first used in section 4.1). Is that a term used in variogram analysis? If so, please describe it in the methods, as I (and I imagine many people) are not very familiar with variograms. I see the term "sill" has been used in section 3.2—is that what the scale break is? If so, please only use one term, define it, and then explain what different values may mean. For example, in Figure 5, what is the significance of the different scale break values?

SMB units: Throughout the manuscript, SMB is reported in units of millimeters (mm). However, SMB is generally reported as a unit of mass change over time such as mm w.e. yr$^{-1}$, m w.e. yr$^{-1}$, or Gt yr$^{-1}$ (Lenaerts et al., 2019). In the manuscript, SMB units of mm should instead be reported as mm w.e. yr$^{-1}$.

**Line-by-line comments**
Comments are numbered as "[page number].[line number]". For example, "1.12" refers to line 12 on page 1.

Abstract

1.12: Change "over next decades" to "over the next decades"

1.12: Change "evolution surface mass loss" to either "evolution of surface mass loss" or "evolving surface mass loss"

1.19: Please also mention the other variables that are assessed and mentioned later in the manuscript (air temperature and surface temperature)

1.21: Specify which variable is being discussed here (SMB?)

Introduction

2.1–2.2: The use of "extension and persistency" is confusing here. I understand what the authors mean by "persistency" but not by "extension." If this refers to the surface melt increasing in strength and duration, consider rewriting this sentence as: "The persistency and intensity of surface melting has also been increasing since 1979, as measured by passive microwave satellite observations [citations]."

2.4: Change "evolution surface mass loss" to either "evolution of surface mass loss" or "evolving surface mass loss"

2.5–2.6: Specify what is meant by "actual mass loss." As compared to what? The authors could specify that remote sensing observations can provide information about surface height changes but are unable to attribute height change to a mass change without more information about snow/firn compaction (e.g., Smith et al., 2023).

2.11–2.16: These statements could benefit from references to specific examples where a finer spatial resolution would have improved results. Several broad examples are mentioned, but citing papers that specifically mention the limitations of the spatial resolution could be helpful.

Datasets

3.6–3.7: Change "Greenland ice sheet" to "GrIS"

3.27: Specify what type of dataset is being referred to in "we used the dataset collected by Machguth". In other words "…the PROMICE dataset…" or "…the SMB dataset…" Though this section (2.3) contains "PROMICE" in its title, nowhere in the text of this section does it say "PROMICE".

4.8: What is meant by "SMB variable"? I thought there were only two model outputs (original and downscaled), but this reads as if there are three.

4.13: Change "Greenland ice sheet" to "GrIS"

Methods

5.3–5.5: Consider rewording these first two sentences for clarity; the phrases in parentheses feel disjointed. Something like: "We adopted the approach used by Noël et al. (2016), in which a statistical downscaling method was applied to RACMO to achieve a 1-km horizontal resolution. Here we use a similar methodology applied to MAR, but instead downscale the product to 100 m horizontal resolution."

5.8: Change "Greenland ice sheet" to "GrIS"

5.15–5.17: The specific description of the pixel and line colors is unnecessary in the text. I suggest either removing these sentences ("The local linear…" and "The dashed red…") or moving them to the figure caption if not already mentioned in the caption.

5.28–5.31: Consider editing this sentence for concision and removing/rewording "embarrassingly parallel problem".

6.2: Consider changing "I/O" to "input/output" to avoid computer science jargon/abbreviations that may be unfamiliar to some.

6.13–6.16: Modify or move this to the Figure 2 caption (see early comment on lines 5.15–5.17).

6.20: Change "constrains" to "constraints"

6.20–6.21: Please expand on this statement. Why was this not necessary in Noël et al. (2016)? What exactly are the physical constraints are how are they applied?

6.23: Is the citation referring to this manuscript? If so, I believe it is unnecessary to add.

6.25: Change "slop" to "slope"
7.1–7.2: Please reword the sentence beginning with "The knowledge of…" I am confused by its meaning.

7.22: Change "th" to "the"

Results and discussion

8.21: Change "remains unvaried, being equal to 2.6 °C" to "remains unvaried at 2.6 °C"

8.24–8.25: Please reword or expand on this sentence in order to clarify the meaning. Specifying the actual physical constraints applied (either here or in the methods as earlier mentioned) could help with clarity and thoroughness.

8.30: I believe this is the first use of "semi-variogram" in the manuscript. How does this differ from just "variogram"? The prefix "semi" is also used in Figure 5 and 6 but not mentioned in the methods section describing variograms. Please either define it or only use "variogram".

8.32–9.1: Please refer to Figure 5 at the end of this sentence, especially since Figure 3 was just mentioned. Additionally, are the numbers reported here meant to match those shown in Figure 5 (13,373, 11,384, and 24,171 km)? If so, the rounded values should be reported as "13.4 km", "11.4 km", and "24.2 km", respectively.

9.12: Is "break scale" correct or should it be "scale break"?

10.1: Remove "from a quantitatively point of view"

11.4: Should "negative" instead be "positive"? Or should the equation be flipped? As it is written, if $RMSE_{100m}$ is smaller than $RMSE_{6km}$ (and thus the downscaled product shows improvement), $\Delta RMSE$ would be positive, not negative. Based on Figure 8 and its caption, I believe the equation should be flipped so its $RMSE_{100m} - RMSE_{6km}$

Conclusions

12.5–12.11: Please reword these sentences since they are very long. Splitting each sentence into two would help.

12.17-19: Reword for clarity.

Figures

Figure 1: Consider changing the northing/easing values to latitude/longitude. This is not a necessity for publication, but would be more helpful for the reader if it is not too much trouble, especially since Table 1, Table 2, and Figure 8 all use lat/lon. Also, if the range of the color bar is adjusted to 0–3200 m, it will show more contrast on the map. As it is now, it all looks like one shade of grey. Summit is at an elevation of ~3200 m, so extending the color bar to 4000+ m is unneeded.

Figure 2: Either change "Latitude" and "Longitude" to "Northing" and "Easting" or report values of lat/lon in panel (a). The caption needs further details and should mention all of the features in the figure itself. The small black dots in (a) and the the blue circles in (a) and (b) need to be described in the caption. The text from the body of the manuscript that describes the blue dots (see earlier comment) could be moved here.

Figure 3: Either change "Latitude" and "Longitude" to "Northing" and "Easting" or report values of lat/lon. The color bar needs to be larger so it's easier to see and should be labeled as "surface temperature". The caption should also specifiy "surface" temperature. Also, where (geographically) is this figure showing? Please either include an inset map of the ice sheet or refer to where it is in the caption. If it is one of the regions in Figure 1, please indicate so in the caption.

**References used in this review**

Lenaerts, J. T., Medley, B., van den Broeke, M. R., & Wouters, B. (2019). Observing and modeling ice sheet surface mass balance. Reviews of Geophysics, 57(2), 376-420. https://doi.org/10.1029/2018RG000622

Smith, B. E., Medley, B., Fettweis, X., Sutterley, T., Alexander, P., Porter, D., & Tedesco, M. (2023). Evaluating Greenland surface-mass-balance and firn-densification data using ICESat-2 altimetry. The Cryosphere, 17(2), 789-808. https://doi.org/10.5194/tc-17-789-2023

---

## Author Comment (AC1)

Reply Review # 1

Reviewer # 1

1  General comments

This manuscript presents a Greenland Surface Mass Balance (SMB) product which was downscaled from 6km to 100m reso-lution using output from the regional climate model MAR and demonstrates that the downscaled dataset exhibits an predom-inantly better agreement with observations than the respective original MAR output at its native resolution (which is already at a very high resolution). To my knowledge the data product is unique in its extremely high resolution and Greenland wide coverage. The presented analysis convincingly demonstrates the improved quality of the SMB data and this work could be a valuable source for the community with respect to small scale applications. The manuscript is clearly structured and most parts are easy to understand, even though some sentences could possibly be decluttered and shortened (examples in the specific comments).

Nevertheless, being interested in downscaling approaches in general rather than in small scale applications, I have some major concerns which mostly concern the general approach.

2  Major comments
 The downscaling approach will be most effective where the MAR topography and the 100m DEM strongly differ and where topographic gradients are large and are dominating the temperature distribution. Towards the coast and on high altitude plateaus the temperature and SMB distribution might be unrelated to elevation. Please provide a map of height difference between the 100m DEM and the native MAR orography, possibly in a supplement.

R: As requested by the reviewer, we are attaching below a figure of the difference between MAR DEM at 6 km and at 100 m. The largest differences, as expected, occur along the coast, where also runoff and temperature are strongly dependent on the elevation. This shows that potential impact of the 100 m spatial resolution vs. the 6 km.

[Figure]

It is not clearly stated, and it should be, where the here applied downscaling approach differs from the one in Noel et al. (2016). An indeed major difference is, that here SMB is downscaled directly (p. 6, l.23), while in Noel et al. (2016) only the SMB components melt, runoff and

sublimation are downscaled while precipitation is interpolated and SMB and refreezing are recalculated from the downscaled components. I am not convinced that downscaling SMB in total is a similarly good choice and would be interested to see the correlation of SMB to elevation (similar to Fig.3 in Noel et al. (2016)).

R: We report below examples of the correlation between SMB and elevation for the whole ice sheet for three different dates, as reported in the figure. As we can observe, there is a mild correlation between SMB values and elevation for relatively low elevation values. However, this is also accompanied by a spread (e.g., large bias) and a saturation after a certain elevation. This is the case for days 150 (May 29) and 250 (September 6). In the case of day 350 (December 15) we find no relationship between the two terms. The dependency for low elevation values might be due to the stronger dependency of the SMB values to runoff. In the case of accumulation, indeed, we do not anticipate any relationship. One aspect that the reviewer points out concerns the direct downscaling of SMB instead of its components. When we did that we found that the performance of the algorithm that was directly downscaling SMB values was better than the one using the sum of the terms downscaled (only runoff and sublimation). Therefore, we decided to downscale directly the SMB values.

[Figure]

Day 150 - 1980

Day 250 - 1980

[Figure]

Day 350 - 1980

Furthermore it should be explained how grid points outside of the 6km ice mask are treated.
*R: We are excluding pixels where the ice sheet covered area is less than 99 %. In practice, we only perform the downscaling over the ice sheet, excluding ocean, tundra, etc.*

I also wonder how much information is actually gained from going to ever increasing resolution (e.g. when going from 6km to 1km to 100m). Is it possible to repeat the SMB downscaling for 1km and compare to stake measurements?

R: We thank the reviewer for this important point. We did perform the downscaling and compared the obtained SMB with measured values, as done in the case of the 100 m. We found that the metrics (e.g., R2, RMSE) for the products at 100 m and 1 km are very similar. Nevertheless, when we computed the spatial autocorrelation of the two products - as done in Figure 5 - we found out that the product at 100 m was able to better match the scale breaks of the measured quantities, pointing to a greater sensitivity to the processes leading to the SMB change.Based on these results, we think the 100 m is a suitable resolution as it doesn't deteriorate the performance at 1 km (similar to what done in Noel et al., 2016) and can better capture spatial variability.

Where the correlation of a variable to elevation is weak, an elevation based downscaling will likely smooth the signal rather than adding finer structure (since regression parameters are interpolated). In these regions I would expect that simple interpola-tion to 100m resolution would produce better results. Therefore it would be interesting to see the same statistics for 6km-SMBs being interpolated to the precise stake location.
R: We are not sure we have properly interpreted the reviewer's request. We suspect they are asking for a comparison between the downscaled results obtained with the method here used and a simple linear interpolation of adjacent pixels. We think this comparison wouldn't be helpful as the linear interpolation would not be able to consider the relationship between temperature change and pixel (or SMB, etc.). We apologize if this was not what the reviewer was referring to.

3 Specific comments

p. 1, l 21: „In the case of the downscaled MAR product": unnecessary repetition. The formulation "in the case of" is heavily used in general and in some case it is redundant or makes the text a bit clumsy.

R: We will remove that portion of the sentence.

p. 1, ll. 23-24: slope and intercept are interchanged.

R: we corrected in the manuscript, thanks

p. 1, l. 28: specify that this study was analyzing North and Central Greenland
R: We added that, thanks

p.2, l.14: maybe provide references for datasets which provide resolution of 100s of meters.

R: We are not familiar with any specific product currently providing mass loss outputs at 100 m . This is the reason why we developed our product. Remote sensing products exist but they mostly look at surface melt extent, duration rather than mass loss.

p.2, ll. 15-16: this is a bit elusive. Can you specify how understanding englacial systems or ocean interaction would benefit from higher degree of detail at the surface (given that mass is conserved with respect to the source data)?
R: We thank the reviewer for this comment. A higher spatial resolution product would allow to better constraint where the water might go when such information is coupled with a digital elevation model. For example, a 6 km product might suggest that for a specific pixel the SMB value would be , let us say, X mmwe but a large portion of this might be geographically located along a specific side (e.g., west or east) with repercussions on where runoff is reaching the ocean.

p.3,l.12: is the TT variable 3-dimensional air temperature or near surface air temperature? Is it possible to specify the height above surface?
R: TT is the temperature at 2m above the surface.

p.4,l.1: typo, pint-> point
R: Corrected, thanks.

p.4,l.11: precise: values of near surface air temperature
R: We corrected that, thanks

p.4,l.24: . . . we use surface temperature fields from seven different. . .
R: Corrected

p.5,l.13-14: It should be stated that this (I guess) is referring to pixels at the margins of the ice masks
R: Thanks, we have added a note specifying this.

p.6,l.20: specify what the physical constraints are in terms of temperature and SMB.
R: We added a sentence explaining that the physical constraint concerns mass conservation for each pixel.

p.6,l.21: typo, constrains – constraints?
R: Corrected , thanks

p.6,l.25: typo, slop->slope
R: Corrected, thanks

section 3.2: I had a hard time reading this section. Maybe concentrate a bit more on what information this analysis provides and how to interpret it. Introduce scale brakes here.
R: Thanks. We have introduced scale breaks when we define the variogram terms. We hope this is sufficient. We have also re-written some of the sentences and we hope this section is now clear.

p.8,l.5: what do you use for the comparison with the original MAR output? Is it nearest neighbor or do you interpolate to the station location or do you interpolate to 100m grid and then choose the nearest neighbor?
R: We use the nearest neighborhood

p.8,l.18: this is unclear to me. What are the sole pixels?
R: We removed that sentence as it is not necessary. Thanks for pointing this out.

p.8,l.24: reword, maybe: the similarity in mean differences is not surprising. . .
R: we rephrased the sentence according to reviewer's suggestion.

p.9, ll.10-12: confusing sentence. Please rephrase.
R: That sentence was not supposed to be there. Apologies. We removed it.

p.9, ll.28-31: confusing, please rephrase. Maybe: Against our expection. . .
R: Thanks. We used "unexpectedly"

p.10, ll.31-33: It needs to be noted that Fettweis et al. (2020) also applied an elevation correction and interpolated to in stake location.
R: For clarifying, we added the following text:
Indeed, as explained in Fettweis et al. (2020), the SMB was extrapolated (interpolated + corrected) to the common 1km grid by applying an elevation gradient as done here. One of the key issues raised by the first SMB model intercomparison performed by Vernon et al. (2013) was the high dependency of modelled integrated SMB values to the ice sheet mask used. To mitigate

this problem, we interpolate all model outputs to the same 1 km grid used in the Ice Sheet Model Intercomparison Project for CMIP6 (ISMIP6). This resolution is chosen because the highest resolution model outputs (e.g. RACMO2.3p2) are available at 1 km and choosing a coarser resolution could compromise their quality. A common grid also allows a comparison on two common ice sheet masks: the contiguous Greenland Ice Sheet, which is common to all the models and the Greenland Ice Sheet plus peripheral ice caps and mountain glaciers, common to all the models except the two PDD models. Unless otherwise indicated, the SMB components have been interpolated to 1 km using a simple linear interpolation metric of the four nearest inverse-distance-weighted model grid cells. Moreover, as done in Le clec'h et al. (2019), the interpolated 1 km SMB and runoff fields have been corrected for elevation differences between the model native topography and the GIMP 250 m topography (upscaled to 1 km here), using time- and space-varying SMB–elevation gradients, similar to Franco et al. (2012) and Noël et al. (2016). No correction was applied to precipitation after interpolation to 1 km.

Figures: please check x and y labels for the maps (distance,longitude, latitude) Fig. 2: m and q are not consistent with Eq. 1 (interchanged)
R: Done, thanks

Fig. 3: check colorbar label
R: Done, thanks

Fig. 8: it would be interesting to also show RMSE by topographic slope.
R: please see below.

[Figure]

[Figure]

---

## Author Comment (AC2)

Manuscript title: A computationally efficient statistically downscaled 100 m resolution Greenland product from the regional climate model MAR

Authors: Marco Tedesco, Paolo Colosio, Xavier Fettweis, Guido Cervone

**General comments**

This study uses a statistical downscaling technique to enhance the horizontal resolution of the Mod.le Atmosph.rique R.gional (MAR) regional climate model and produce an improved surface mass balance (SMB) product. The authors use an impressive breadth of datasets and tools both for statistically downscaling and for evaluating the final product. The product itself is clearly an improvement, is of notably high spatial resolution, and likely has many valuable applications. The high resolution over such a large area is computationally remarkable. The manuscript is well-organized and the flow of ideas is very logical.

My main concern is the final step of the methodology, which leaves the reader wondering how well the statistical downscaling works without applying physical constraints. I have also described two minor comments and several line-by-line comments that are mostly concerned with improving the writing itself. Some clarity of the research is lost due to longer/confusing sentences, so I have suggested some improvements below.

**Major comments**

Description of final step of methodology: There is an insufficient description of the "physical constraints" applied in the final step of the downscaling in section 3.1 This needs further elaboration, especially since it differs from No.l et al. (2016) and is later referred to in section 4.1. The authors should

(1) provide a reason as to why the mass conservation issues did not arise in No.l et al. (2016)

R: We thank the reviewer for this comment. We are not sure, however, how to answer to the first question as we did not author the paper in object and are not sure neither why mass conservation was not considered nor why this is not discussed in the paper. As the reviewer can imagine, we think this is an important step of the methodology because it assures that the downscaling (which is very simple in its own nature) does not alter the outputs of the MAR model. The coefficients used for the downscaling , indeed, use such values and we want to preserve consistency between the two datasets (coarse and fine) so tha there is also compatibility between the two products. After all, the downscaling technique is not supposed to introduce any further information to the computed values (e.g., differently of how it would be in the case, for example, of a machine learning model or any dynamical model that accounts for changes in the state of the system). Rather, the goal is to provide a downscaled product from the MAR model outputs.  To reiterate, we are not sure why this issue was not raised in the paper by Noel et al. but we think it is important.

, (2) describe exactly what these physical constraints are,

There are two physical constraints: the first constraint is on temperature. The downscaled outputs of the temperature are such that the mean average temperature of the pixels at the finer spatial scale is equal to the temperature of the original MAR pixel at 6 km; the second constraint is on the SMB. In this case, the sum of the SMB values at finer spatial scales is set to equal the SMB value for the coarser corresponding MAR pixel.

and (3)

report on how they affect the final product. Without any additional information, the application of the physical constraints could be interpreted as forcing the final product to fit within the expectations. As this is likely not the case, a description of these steps will give the reader more confidence in the methodology.

R: We ran the downscaling previously without applying the physical constraints (which are now better explicit in the paper) and found small differences between the two products (<0.1 %).

**Minor comments**

Scale break: I am unsure of the meaning of "scale break" (first used in section 4.1). Is that a term used in variogram analysis? If so, please describe it in the methods, as I (and I imagine many people) are not very familiar with variograms. I see the term "sill" has been used in section 3.2—is that what the scale break is? If so, please only use one term, define it, and then explain what different values may mean. For example, in Figure 5, what is the significance of the different scale break values?

R: thank you for the suggestion. We introduced the term "scale break' when we present the variogram in Section 3. The scale break is, indeed, the spatial scale at which the autocorrelation changes, indicating that different processes might be dominating (e.g., large scale atmospheric processes vs. local wind effects). We point that the *range* can be seen as a scale break but there can also be several scale breaks before the sill is reached, depending on the drivers controlling the modeled process. We added this in the manuscript and thank the reviewer for the comment.

SMB units: Throughout the manuscript, SMB is reported in units of millimeters (mm). However, SMB is generally reported as a unit of mass change over time such as mm w.e. yr-1, m w.e. yr-1, or Gt yr-1 (Lenaerts et al., 2019). In the manuscript, SMB units of mm should instead be reported as mm w.e. Yr-1.

R: Done.

**Line-by-line comments**

Comments are numbered as "[page number].[line number]". For example, "1.12" refers to line 12 on page 1.

Abstract

1.12: Change "over next decades" to "over the next decades"

R: Done , thanks

1.12: Change "evolution surface mass loss" to either "evolution of surface mass loss" or "evolving surface mass loss"

R: Done, thanks

1.19: Please also mention the other variables that are assessed and mentioned later in the manuscript (air temperature and surface temperature)

R: Done, thank you

1.21: Specify which variable is being discussed here (SMB?)

R: Yes, we changed the sentence based on another reviewer's comment. We hope the new version clarifies that we are referring to SMB

Introduction
2.1–2.2: The use of "extension and persistency" is confusing here. I understand what the authors mean by "persistency" but not by "extension." If this refers to the surface melt increasing in strength and duration, consider rewriting this sentence as: "The persistency and intensity of surface melting has also been increasing since 1979, as measured by passive microwave satellite observations [citations]."
R: Thanks, we changed that

2.4: Change "evolution surface mass loss" to either "evolution of surface mass loss" or "evolving surface mass loss"
R: Done, thanks

2.5–2.6: Specify what is meant by "actual mass loss." As compared to what? The authors could specify that remote sensing observations can provide information about surface height changes but are unable to attribute height change to a mass change without more information about snow/firn compaction (e.g., Smith et al., 2023).
R: We changed that, thanks !

2.11–2.16: These statements could benefit from references to specific examples where a finer spatial resolution would have improved results. Several broad examples are mentioned, but citing papers that specifically mention the limitations of the spatial resolution could be helpful.

R: Thanks, We have rewritten and modified those sentences.

Datasets
3.6–3.7: Change "Greenland ice sheet" to "GrIS"
R: Done. We also changed it on other occurrences.

3.27: Specify what type of dataset is being referred to in "we used the dataset collected by Machguth". In other words "…the PROMICE dataset…" or "…the SMB dataset…" Though this section (2.3) contains "PROMICE" in its title, nowhere in the text of this section does it say "PROMICE".

R: We added "SMB" to specify that is the same dataset

4.8: What is meant by "SMB variable"? I thought there were only two model outputs (original and downscaled), but this reads as if there are three.
R: apologies, we removed the word "variable". There are indeed two SMB modeled outptus.

4.13: Change "Greenland ice sheet" to "GrIS"
R: Done.

Methods
3
5.3–5.5: Consider rewording these first two sentences for clarity; the phrases in parentheses feel disjointed. Something like: "We adopted the approach used by No.l et al. (2016), in which a

statistical downscaling method was applied to RACMO to achieve a 1-km horizontal resolution. Here we use a similar methodology applied to MAR, but instead downscale the product to 100 m horizontal resolution."

R: Thanks , we did that

5.8: Change "Greenland ice sheet" to "GrIS"
R: Done

5.15–5.17: The specific description of the pixel and line colors is unnecessary in the text. I suggest either removing these sentences ("The local linear…" and "The dashed red…") or moving them to the figure caption if not already mentioned in the caption.
R: We removed it , thanks.

5.28–5.31: Consider editing this sentence for concision and removing/rewording "embarrassingly parallel problem".
R: thanks we removed that portion and rewrittent the sentence.

6.2: Consider changing "I/O" to "input/output" to avoid computer science jargon/abbreviations that may be unfamiliar to some.
R: Done.

6.13–6.16: Modify or move this to the Figure 2 caption (see early comment on lines 5.15–5.17).
R: We moved the text to the caption.

6.20: Change "constrains" to "constraints"
R: Done

6.20–6.21: Please expand on this statement. Why was this not necessary in No.l et al. (2016)? What exactly are the physical constraints are how are they applied?

R: As we explained to the other reviewer, We are not sure how to answer the first question as we did not author the paper in object and are not sure why mass conservation was not considered. There are two physical constraints: the first constraint is on temperature. The downscaled outputs of the temperature are such that the mean average temperature of the pixels at the finer spatial scale is equal to the temperature of the original MAR pixel at 6 km; the second constraint is on the SMB. In this case, the sum of the SMB values at finer spatial scales is set to equal the SMB value for the coarser corresponding MAR pixel.

6.23: Is the citation referring to this manuscript? If so, I believe it is unnecessary to add.
R: That citation was a mistake. Thanks for pointing this out

6.25: Change "slop" to "slope"
R: Done

7.1–7.2: Please reword the sentence beginning with "The knowledge of…" I am confused by its Meaning.
R: Apologies, we rewrote that sentence.

7.22: Change "th" to "the"
R: Done

Results and discussion
8.21: Change "remains unvaried, being equal to 2.6 Åã C" to "remains unvaried at 2.6 Åã C"
R: Done

8.24–8.25: Please reword or expand on this sentence in order to clarify the meaning. Specifying the actual physical constraints applied (either here or in the methods as earlier mentioned) could help with clarity and thoroughness.
R: We modified that sentence following another reviewer's suggestion.

8.30: I believe this is the first use of "semi-variogram" in the manuscript. How does this differ from just "variogram"? The prefix "semi" is also used in Figure 5 and 6 but not mentioned in the methods section describing variograms. Please either define it or only use "variogram".
R: thanks, we will be consistent.

8.32–9.1: Please refer to Figure 5 at the end of this sentence, especially since Figure 3 was just mentioned. Additionally, are the numbers reported here meant to match those shown in Figure 5 (13,373, 11,384, and 24,171 km)? If so, the rounded values should be reported as "13.4 km", "11.4 km", and "24.2 km", respectively.

R: Done, thanks.

9.12: Is "break scale" correct or should it be "scale break"?
R: scale break is the appropriate version, thanks

10.1: Remove "from a quantitatively point of view"
R: Done

11.4: Should "negative" instead be "positive"? Or should the equation be flipped? As it is written, if $RMSE_{100m}$ is smaller than $RMSE_{6km}$ (and thus the downscaled product shows improvement), $\Delta RMSE$ would be positive, not negative. Based on Figure 8 and its caption, I believe the equation should be flipped so its $RMSE_{100m} - RMSE_{6km}$

R: Yes, thanks for noticing the typo, the equation is $\Delta RMSE = RMSE_{100m} - RMSE_{6km}$.

Conclusions
12.5–12.11: Please reword these sentences since they are very long. Splitting each sentence into two would help.

R: thanks, We shortened and reworded that sentence.

12.17-19: Reword for clarity.
R: Done, thanks

Figures
Figure 1: Consider changing the northing/easing values to latitude/longitude. This is not a

necessity for publication, but would be more helpful for the reader if it is not too much trouble, especially since Table 1, Table 2, and Figure 8 all use lat/lon. Also, if the range of the color bar is adjusted to 0–3200 m, it will show more contrast on the map. As it is now, it all looks like one shade of grey. Summit is at an elevation of ~3200 m, so extending the color bar to 4000+ m is Unneeded.

R: Thanks. We replaced the figure.

Figure 2: Either change "Latitude" and "Longitude" to "Northing" and "Easting" or report values of lat/lon in panel (a). The caption needs further details and should mention all of the features in the figure itself. The small black dots in (a) and the the blue circles in (a) and (b) need to be described in the caption. The text from the body of the manuscript that describes the blue dots (see earlier comment) could be moved here.

R: Done

Figure 3: Either change "Latitude" and "Longitude" to "Northing" and "Easting" or report values of lat/lon. The color bar needs to be larger so it's easier to see and should be labeled as "surface temperature". The caption should also specifiy "surface" temperature. Also, where (geographically) is this figure showing? Please either include an inset map of the ice sheet or refer to where it is in the caption. If it is one of the regions in Figure 1, please indicate so in the Caption.

R: Done.

References used in this review
Lenaerts, J. T., Medley, B., van den Broeke, M. R., & Wouters, B. (2019). Observing and modeling ice sheet surface mass balance. Reviews of Geophysics, 57(2), 376-420. https://doi.org/10.1029/2018RG000622
Smith, B. E., Medley, B., Fettweis, X., Sutterley, T., Alexander, P., Porter, D., & Tedesco, M. (2023). Evaluating Greenland surface-mass-balance and firn-densification data using ICESat-2 altimetry. The Cryosphere, 17(2), 789-808. https://doi.org/10.5194/tc-17-789-2023